# A Functional Genomics Review of Non-Small-Cell Lung Cancer in Never Smokers

**DOI:** 10.3390/ijms241713314

**Published:** 2023-08-28

**Authors:** Mohammad Hamouz, Raneem Y. Hammouz, Muhammad Ahmed Bajwa, Abdelrahman Waleed Alsayed, Magdalena Orzechowska, Andrzej K. Bednarek

**Affiliations:** Department of Molecular Carcinogenesis, Medical University of Lodz, Żeligowskiego 7/9, 90-752 Lodz, Poland; mhamouz879@gmail.com (M.H.); muhammad.abajwa@gmail.com (M.A.B.); awalsayedd@gmail.com (A.W.A.); magdalena.orzechowska@umed.lodz.pl (M.O.); andrzej.bednarek@umed.lodz.pl (A.K.B.)

**Keywords:** non-small-cell lung cancer, never smokers, genomic landscape, tumour microenvironment, targeted therapy, immunotherapy, gender

## Abstract

There is currently a dearth of information regarding lung cancer in never smokers (LCINS). Additionally, there is a difference in somatic mutations, tumour mutational burden, and chromosomal aberrations between smokers and never smokers (NS), insinuating a different disease entity in LCINS. A better understanding of actionable driver alterations prevalent in LCINS and the genomic landscape will contribute to identifying new molecular targets of relevance for NS that will drastically improve outcomes. Differences in treatment outcomes between NS and smokers, as well as sexes, with NSCLC suggest unique tumour characteristics. Epidermal growth factor receptor (EGFR) tyrosine kinase mutations and echinoderm microtubule-associated protein-like 4 anaplastic lymphoma kinase (EML4-ALK) gene rearrangements are more common in NS and have been associated with chemotherapy resistance. Moreover, NS are less likely to benefit from immune mediators including PD-L1. Unravelling the genomic and epigenomic underpinnings of LCINS will aid in the development of not only novel targeted therapies but also more refined approaches. This review encompasses driver genes and pathways involved in the pathogenesis of LCINS and a deeper exploration of the genomic landscape and tumour microenvironment. We highlight the dire need to define the genetic and environmental aspects entailing the development of lung cancer in NS.

## 1. Introduction

Although the smoking population is decreasing, lung cancer is amongst the most common cancers and the leading cause of cancer-related deaths. Due to a decrease in the number of tobacco smokers, the proportion of never smokers is increasing, yet approximately 10–25% of lung cancer cases occur in never smokers [1]. Moreover, worldwide, lung cancer in never smokers (LCINS), specifically, is the seventh leading cause of cancer-related deaths [2]. The first and most visible difference is the significantly lower frequency of mutations in non-smokers, which indicates both a different aetiology and a different progression of this cancer. Unfortunately, clinical features of never smoker adenocarcinomas include more frequent pleural metastasis 46% compared to 25% in smokers, along with differences in mutational profiles and in demographics, being prominent with increased risk in females [3,4], as well as better prolonged and overall survival [5,6]. Epidemiology shows that, as women smoke more, with the increased association between tobacco marketing strategy and women’s emancipation [7], the incidence of lung cancer increases. In Poland (according to the Polish National Cancer Registry), lung cancer deaths among women increased from 4500 cases to over 8000 between 2001 and 2019; at the same time, the number of deaths among men fell from about 17,000 to 15,000 (onkologia.org.pl, accessed on 1 June 2023). This may have been due to an increase in smoking among women, especially trendy light-flavoured cigarettes.

Data indicate that LCINS is a unique tumour with different biology and tumour microenvironment (TME) [6], featuring an adenocarcinoma-predominant histology [8] compared to lung cancer in smokers [6]. In fact, the TME in never smokers (NS) is less compromised than that in current smokers (CS) with alterations in 17 out of 20 immune response-related pathways in lung adenocarcinoma (LUAD) in different smoking groups [6]. Additionally, there is a different composition of immune cells between CS and NS, with more resting mast cells and resting CD4+ memory T cells in NS, which are associated with better outcomes, compared to activated mast and CD4+ memory T cells in CS [6]. Furthermore, according to the Sherlock-Lung study, LUAD smokers are usually classified into subtypes comprising histological features, as well as transcriptomic, epigenomic, and genomic changes, whereas this is yet to be uncovered in NS [8]. We previously identified cell-cycle-related genes showing elevated expression in NS, associated with mismatch repair, homologous recombination, DNA replication, VEGF signalling pathway, T-cell receptor signalling pathway, ErbB signalling pathway, and GnRH signalling pathway. On the other hand, elevated expression of CS-related genes was more specific to the regulation of cell-cycle processes. We also identified the most frequently and statistically significant mutations between NS and CS from TCGA cohort from cBioportal [9]. Newly analysed data regarding top mutations across patients from never smokers, current smokers, and reformed smokers are presented in Table 1. Of relevance to this paper and further discussed in upcoming sections are the *EGFR*, *TP53*, *KRAS* and *ERBB2* mutations.

General risk factors associated with lung cancer include age, exposure to tobacco smoke (ETS), ionising radiation, radon gas, inherited genetics, hormone replacement therapy, and pre-existing lung diseases including chronic obstructive pulmonary disease (COPD) [3,6]; however, most LCINS patients have no clearly defined risk factors [8]. The clinical management of this disease remains a major challenge as its pathogenic mechanism is complex, let alone the dearth of information regarding NS. Since frequent driver mutations in LCINS are different to those occurring in smokers, this different oncogenic activating alteration consequently makes LCINS a separate disease entity. Furthermore, from observing *TP53*, *KRAS*, and *EGFR* mutational patterns and frequencies between CS and NS, it is evident that LUAD arises via different pathogenic pathways [10]. Indeed, in LUAD these frequent driver mutations were detected in 95% of NS patients, of which 78% were treatable with highly specific targeted therapies, compared to 65% in CS, of which only 47% were actionable [11]. Additionally, to add to the complexity, there have been several treatment strategies with discrepancies in responses according to demographic factors. For instance, between CS and NS, erlotinib, an epidermal growth factor receptor tyrosine kinase inhibitor (EGFR-TKI), showed notable responses in LUAD female NS of Asian ethnicity [10].

It is thought that certain immunological features, irrespective of tobacco smoking, contribute to lung cancer development. In an ideal healthy tissue, the immune system recognises and destroys malignant cells. However, tumours evolve mechanisms to evade host immune-mediated surveillance, including the expansion of a local immunosuppressive microenvironment, induction of dysfunctional T-cell signalling, and upregulation of inhibitory immune checkpoints. This has led to the introduction of immune checkpoint blockade (ICB), which reactivates the intrinsic anti-tumour immune response via blocking inhibitory immune receptors expressed on the surface of cancer cells of immune cells within the TME. Immune checkpoint inhibitors (ICIs) actively target the compromised milieu rather than the tumour itself. Nevertheless, durable responses to ICI have not been successful in all cancers, as a number of cancers were more efficiently ‘hidden’ from host immune surveillance than others, or so-called immune ‘silent/cold’. A number of signalling pathways are being examined as therapeutic targets; however, both conventional and targeted drugs have unfortunately presented significant adverse effects [12]. In fact, current or former NSCLC smokers receiving anti-PD-1 therapy achieved a response rate of 22.5%, almost double that in NS (10.3%). The better ICB efficacy was attributed to increased carcinogen-induced tumour mutational burden (TMB) [13]. These findings accentuate the urgent requirement for more effective personalised therapeutic strategies. Since this negatively influences the patients’ quality of life and clinical outcome, it calls for focused analysis of molecular carcinogenesis in LCINS, which might even provide insights for therapeutic implications for current, reformed, and never smokers. Additionally, the driver mutations in LCINS could provide roadmaps for precision drug therapies.

In this review, we focus on the current molecular biology and functional genomics of NSCLC and its development in never smokers. We review the current literature regarding key signalling pathways and driver mutations, and we highlight changes in the immune environment, as well as potential therapeutic inhibitors based on pre-clinical and clinical trials. We aim to explore how certain genes that partake in important carcinogenic pathways are involved in LCINS.

## 2. The Genomic Profile of Lung Cancer in Never Smokers

Genome-wide studies (GWS) have clearly indicated that the underlying tumour biology of NS differs strikingly from that of smokers (CS). Distinct pathways are altered between NS and CS [6,9,14], with NS patients having better outcomes [5,6]. A summarised list of some key genes (drivers) in NS NSCLC patients is presented in Table 2.

LCINS features a different pattern of molecular alterations; ergo, it is considered a biologically separate disease entity. Although tobacco smoke is responsible for the majority of lung cancer cases, around 10–20% of all lung cancer diagnoses occur in NS. Exposure to environmental tobacco smoke (ETS) has already been supported and recognised to be the causative agent of lung carcinogenesis in NS; however, some lung cancer patients have not been exposed to these carcinogens, implying that other risk factors/causative agents are yet to be established [19]. LUAD is the most prevalent subtype of NSCLC, which frequently arises among female NS. It adopts a histologically glandular pattern with activating mutations affecting driver genes including *KRAS*, *EGFR*, *BRAF*, and *ALK* fusions, amongst other genetic alterations [20]. We previously identified several NS differentially expressed genes (DEGs) in LUAD, which were involved in cell cycle, VEGF, ErbB, GnRH, and TGF-β signalling pathways, as well as asthma, indicating the role of inflammation in NS [9].

Passive smokers show a similar mutation profile to smokers, albeit with a lower mutational rate overall. Therefore, analysis of the genomic differences between NS and CS will aid in uncovering the cellular and molecular pathways of malignant transformation. Indeed, although NS patients have a lower number of mutations compared to CS, they seem to be conducive for the malignant transformation, whereas, in CS, the numerous mutations seem to be mostly passenger mutations [21]. Even so, there are specific changes that occur in the TME and distinct driver genes, as well as genetic pathway alterations, in NS. In spite of that, NS patients are usually younger, have a better prognosis, and respond better to treatment than smokers, due to the occurrence of certain molecular subtypes [6] including EGFR-TK domain mutations. Verily, EGFR-TK domain mutations were found to be the first statistically significant molecular changes to occur particularly in NS. Furthermore, they are more frequent in NS than in long-term smokers (51% versus 10%) and in LUAD, contrary to cancers of other histologies [22].

The transformation from LUAD to small-cell lung cancer (SCLC) as an outcome of EGFR-TKI resistance has recently been of much interest. This is not as common in NSCLC patients without *EGFR* mutations. The inactivation of *RB1* increases the expression of neuroendocrine markers and decreases the expression of *EGFR*, which is usually detected in the transformed tissue. Furthermore, in 21 patients, this transformation following EGFR-TKI resistance required an inactivation of both *RB1* and *TP53* just before branching. Inactivating gene alterations of both *RB1* and *TP53* could possibly serve as predictive markers for the transformation, since it is common and seems to be a necessary event for both genes to be inactivated in all small-cell lung cancer SCC tissues [15].

Zhang and colleagues conducted an in-depth genomic and mutational signature analysis (WGS) with the hope of providing a guide to the development of precise clinical treatments to benefit LCINS. They recruited 232 NSLSC NS patients and established three genetic subtypes unique to NS. TMB was almost sevenfold lower in NS than in smokers [1,21] and significantly associated with tumour stage, histology, and age, but not tumour purity. A higher frequency of *EGFR* mutations was found in females than in males. Additionally, EGFR signalling was increased due to *EGFR* or *KRAS* mutations, and mutations in *RBM10* and *TP53* were found to frequently co-occur with *EGFR*. Moreover, co-occurring patterns were also found between *RBM10* and *PIK3CA*, as well as between *TP53* and *ERBB2* and marginally significant enrichment of *SETD2* mutations, in samples with oncogene fusions, particularly in TP53-proficient tumours. A strong mutually exclusive distribution was observed across the genes in the RTK–Ras pathway which were altered in a total of 54.3% tumours. *EGFR* was the most frequently altered, followed by *KRAS*, *ALK*, *MET*, *ERBB2*, *ROS1*, and *RET* [1]. 

Moreover, unsupervised clustering of arm-level somatic copy number alterations in the Sherlock-Lung study identified three distinct subtypes. Briefly, subtype 1 called ‘piano’ tumours had the least number of mutations growing slowly over the years, with the most frequently mutated gene being *KRAS*. Oncogenic *KRAS* mutations were involved in LUAD since they induced proliferation of bronchioalveolar stem cells. Subtype 2 ‘mezzo-forte’ tumours demonstrated mutations in *EGFR*, a common mutation found in lung cancers exhibiting faster tumour growth. ‘Mezzo-forte’ was enriched with chromosome arm-level amplifications of 1q, 5p, 7p/q and 8q. Lastly, they identified the ‘forte’ subtype which grew the quickest, was most similar to lung cancers among smokers, and was dominated by whole-genome doubling (WGD). Forte also had a low TMB and a larger proportion of subclonal mutations indicative of extensive intra-tumour heterogeneity. Piano also had a small number of known driver gene mutations suggesting stem-like features, in addition to a new driver gene *UBA1*, which acts as one of the main orchestrators of cellular DNA damage response. It is worth mentioning that they did not find major differences in the mutational signature or types between passive and non-passive smokers. However, they observed a few tumours with diesel exhaust signatures. Since smoking-related mutations from passive smokers were below the detection threshold of 15%, it is possible that second-hand tobacco smoke may also act through alternative tumourigenic processes and selective constraints. They also found alterations in *KRAS*, *UBA1*, *RET*, and *ARID1A* to be mutually exclusive in piano. *KRAS* and *UBA1* are important hematopoietic and pluripotent stem-cell regulators, and *RET* is involved in murine hematopoietic stem-cell regulation. *ARID1A* could promote exit from a quiescent cell state, causing high inter-tumoural heterogeneity; thus, it could drive some of the tumours with no detected known cancer driver gene mutations or fusions. On the contrary, the forte and mezzo-forte subtypes were generally clonal with driver gene mutations and WGD or gross somatic copy number alterations (SCNAs) facilitating identification and possibly successful treatment [1]. 

A recent comprehensive genomic and transcriptomic analysis (WES) of 160 tumour and normal LUAD samples from NS highlighted a number of prevalent clinically relevant alterations, allowing a better understanding of the risk factors involved. They categorised NS LUAD into relatively immune cold and hot subtypes. The immune cold subtype tumours, in comparison with immunologically hot tumours, appeared to lack expression of immune markers, such as PD-L1, but were depleted with immune cells. This suggests immune evasion mechanisms that are not very well characterised. Indeed, compared to smokers, NS LUAD tumours had relatively lower TMBs and a higher frequency of mutations in genes such as *CTNNB1*. *CTNNB1* participates in wingless WNT signalling, and WNT signalling activation facilitates immune evasion and contributes to immunotherapy resistance, possibly providing an explanation for the relatively lower response rates associated with immunotherapy targeting PD-L1 in NS LUAD patients compared with smokers. Moreover, they identified a small subset of the samples from NS to have pathogenic and likely pathogenic germline alterations in DNA repair genes. DNA repair genes exclusively mutated in NS included *BRCA1*, *BRCA2*, *FANCG*, *FANCM*, *MSH6*, and *POLD1*. Furthermore, some tumour cells exhibited mutational signatures characteristic of ETS. In fact, they found the genomic features of smokers and NS to be comparable with the activation of RTK/RAS/RAF signalling, a hallmark feature of LUAD. Nevertheless, NS LUAD demonstrated a higher prevalence of targetable driver alterations in RTK/RAS/RAF signalling pathway than smokers [3].

Likewise, the genetic pathway of driver mutations in NS lung cancer are different to those in smokers. An evaluation of therapeutically targetable mutations such as *EGFR*, *ALK*, and *KRAS*, as well as chromosomal rearrangement and fusion of *EML4*, identified increasing odds of presenting these mutations in adenocarcinoma and NS compared to NSCLC and smokers, respectively. Additionally, the mutations of *EGFR* were more prevalent in Asian women in comparison to women of Caucasian/mixed ethnicity [16,23], further highlighting ethnicity as a risk factor. In support of this was a recent Australian study evaluating the risk factors in LCINS. They explored demographic, lifestyle, and health-related factors in NS and found growing evidence that ethnicity could be considered when assessing potential risk factors in LCINS [24]. Furthermore, methylation profiles of LCINS are different from smokers, with 16p chromosomal aberration gain being more frequent in NS [25]. 

A protogenomic study in Taiwan focused on exploring differences in NS LUAD, with a cohort of 83% non-smokers. They included the matched early-stage tumours and their normal adjacent tissues. Their study revealed different mutational profiles of previously explored driver genes in NS with significantly different mutational frequencies. Moreover, in their NS cohort, genes including *EGFR*, *RBM10*, and *RNF213* were amid the top-ranking mutations. Other somatic mutations prevalent in the NS cohort included *ATP2B3* and *TET2*. *RBM10* is an LUAD tumour suppressor, and its loss, due to mutations, could impact its interactions and lead to impaired RNA splicing. Moreover, they observed co-regulated phosphorylation of MAPK pathway proteins, which distinguished patients with high activation to associate with *EGFR* and *KRAS* mutations, while low activation with *TP53* mutations, especially in later stages. A characteristic observation in this study is that there were no significant differences in C>A transversions between smokers and NS, suggesting that other factors aside from smoking contribute to the genomic landscape of NS. C>A transversions were significant in the smoker cohort and C>T transitions were significant in the NS cohort. Around 85% of the patients had *EGFR* mutations followed by *TP53* (33%) and *RBM10* (20%) [16]. In a previous study, we also identified *EGFR* mutations to be statistically significant (*p* < 0.005) between CS and NS, with prevalence in NS [9].

A study by Paik et al. attempted to explain the survival advantage that NS with LUAD with stage IIIB/IV have over former/current smokers, living 50% longer. In line with previous studies, they also found that NS had a significantly higher proportion of *EGFR* mutations and *ALK* rearrangements, whereas smokers had a higher proportion of *KRAS* mutations. Notably, they did not observe significant overall survival (OS) differences in both groups with identical genotypes. The authors concluded that both smoking groups are not homogeneous, with each group’s individuals having a set of disparate mutations that additively generate an overall prognosis. They went on to note that, although NS have a higher frequency of *EGFR* mutations, EGFR-TKIs should not be the immediate route of treatment. This is because, following treatment with EGFR-TKIs, smokers with *EGFR* mutations exhibit similar survival outcomes to NS. Instead, regardless of smoking history, LUAD patients should initially undergo testing for *EGFR* mutations and rearrangements in *ALK* in an effort to match patients with appropriate targeted therapy, while those who do not harbour mutations in *EGFR/KRAS* or rearrangements in *ALK* should be stratified by smoking history [17].

Hormonal factors are another risk factor in lung carcinogenesis. The link between driver oncogenes and hormonal receptors in LUAD was supported by a study conducted by Mazieres and colleagues examining 140 females with LUAD, which included 63 never smokers. They found female NS to be characterised by older age and a higher frequency of lepidic features compared to smokers. Additionally, they observed differential genetic alterations to be prevalent in NS, including a higher mutation frequency of *EGFR* but a lower frequency of *KRAS* and a higher percentage of oestrogen receptor alpha (ERα). Furthermore, ERα expression correlates with the presence of *EGFR* mutations, associated with both mutational and hormonal biomarkers [18]. This suggests that hormonal factors may be part of the alteration events in NS, especially since females are overrepresented in LUAD never smokers [3], and since we also previously showed that sex-specific changes present an association with cancer progression and prognosis [26]. Hence, exploring both hormonal factors and genetic abnormalities in NS could possibly be therapeutical targets [18].

## 3. Immunological Changes: Tumour Microenvironment in Never Smokers

Although not widely studied, the TME in NS is distinctly different from that of smokers. Moreover, the immune system plays a vital role in cancer development progression. It is thought that specific immunological features contribute to lung cancer development irrespectively of tobacco smoking. When NS are exposed to ETS, immune cells are initially recruited to minimise the damage by carcinogenic substances. Signalling cascades including MAPK, which bridges the switch from extracellular signals to intracellular responses, are involved in not only cell proliferation but also immune escape, contributing to cancer progression [27]. However, when a tumour arises, it might also cause harmful pro-inflammatory and immune reactions, and partake in the harmful TME, contributing to tumour growth invasion and metastatic spread. Therefore, the immune system could protect against cancer progression or enhance tumour growth by influencing the TME and weakening the surrounding normal cells [6]. Moreover, cancer and autoimmune disorders are frequently encountered in elderly patients possibly due to the ageing process, which could also affect changes in innate and adaptive immune function, i.e., immunosenescence. Immunosenescence also causes dysfunctional maturation and function of natural killer (NK) cells and insufficient neutrophil migration, probably due to increased constitutive PI3K activation. In turn, this decline in NK comprises a slower response to inflammatory conditions and affects the adaptive immune system via immunosenescence, altering the function of B and T lymphocytes [28].

An integrative analysis study conducted by Li and colleagues included 11 lung cancer gene-expression datasets that provided data from 1111 LUAD patients and an adjacent 200 samples of normal tissue. They found distinct pathways altered between smokers and NS, with NS having a better outcome. In addition, the transforming growth factor beta (TGFβ) pathway has been identified to contribute to immune dysfunction and to be associated with immune checkpoint inhibitor (ICI) resistance. They identified the *TGFBR2* mutation to predict immunotherapeutic resistance, associated with increased JAK/STAT signalling and immune checkpoints including *CD274*, *LAG3*, *TIGIT*, *PDCD1*, and *PDCD1LG2* [14]. They also characterised the compositional patterns of 21 types of LUAD immune cells. Their study revealed complex and multi-layered associations between the composition of immune cell subtypes and clinical outcome among the smoking groups. In particular they found mast cells and CD4+ memory T cells with completely opposite associations with outcomes in resting and activated status. The number of resting mast cells, not having undergone degranulation, was found to be reduced in tumour samples, in comparison to the adjacent normal tissue and a predictor of favourable outcome. On the other hand, macrophages, activated mast cells following degranulation, and activated CD4+ memory T cells were enriched in the tumour samples, predicting a poor prognosis. NS had more resting mast and CD4+ memory T cells associated with a better outcome, whereas, in smokers, there were more activated mast cells and CD4+ cells, which correlated with a generally worse prognosis [14].

In addition, oxidative stress can be a causative factor for lung carcinogenesis in NS due to the constant exposure to ambient air pollution. Ito et al. investigated the impact of oxidative stress in NSCLC patients who underwent surgical resection including 34 CS and 27 NS by examining oxidative damage on DNA. Immunohistochemistry was used to assess the oxidative damage by examining the accumulation of thymidine glycol (TG). TG is a specific marker for oxidative DNA damage since thymidine is not incorporated into RNA. The mean TG positive rate, indicative of oxidative DNA damage, was significantly higher in smokers compared to NS, and significantly higher in NS than surgical patients with benign lung disease. They also investigated the serum oxidative stress and antioxidant capacity (AOC). The mean level of AOC was found to be significantly lower in NS compared to smokers. Hence, the comparatively low antioxidant potential for NS could be a contributing factor to excessive oxidative DNA damage in lung tissue [28].

Another study aimed to quantify the differences between smokers and NS LUAD patients by analysing immune infiltration and stemness amongst other variables. Owing to its crucial role in lung cancer outcomes, therapies against *NTS* reduce tumour growth and metastasis; an important finding is that *NNAT* and *NTS* were upregulated in NS. Additionally, their overexpression in the NS cohort could be involved in LUAD development. Moreover, they identified mast cells, M2 macrophages, memory resting CD4 + T cells, and dendritic cells to be upregulated in NS, and both *TFF2* and *REG4* were downregulated in NS LUAD. *TFF2* is required by lung macrophages to promote epithelial proliferation; thus, its downregulation in NS could explain the weaker repair capacity and tumour development. On the other hand, *REG4* plays a role in *KRAS* driven lung cancer pathogenesis. They also found a significant difference in the expression for programmed death-ligand 1 (PD-L1) between smokers and NS [29].

Lastly, the mRNA expression-based stemness indices (mRNAsi) showed significant differences among the groups, with NS or reformed smokers exhibiting lower stemness than current and recently reformed smokers. All results provide an understanding of the causes of oncogenesis in NS LUAD and possible therapeutic approaches [29]. This was also supported by an integrated multi-omics study exploring the possible underlying molecular mechanisms among NS, former smokers, and current smokers. In addition to tumour cell stemness, they found different immune content, genome stability, and sensitivity to chemotherapy drugs. Their results also indicated that NS had better OS and disease-specific survival (DSS) than smokers but were substantially more sensitive to multiple chemotherapeutic drugs than smokers. Additionally, leukocyte infiltration, intertumoural heterogeneity, and neoantigen levels were significantly higher in smokers compared to NS [30].

## 4. Abnormalities in Growth-Stimulatory Signalling Pathways

Several oncogenic proteins are either members of cytoplasmic signalling cascades or interact with them, leading to transformations as a result of this deregulation. Recent studies have focused on investigating the carcinogenesis of LCINS. Inarguably, the functions of several pathways with major components are altered in LCINS, making it a separate entity. Some recent studies identified a list of DEGs in female NS LUAD patients enriched in p53, TGF-beta, and cell-cycle signalling pathways [31] and nuclear division pathways [32]. In NSCLC, several signalling pathways have been heavily implicated in both tumourigenesis and progression of the disease. Various specific inhibitors of PI3K, Akt, and mTOR are currently under development for NSCLC, at various stages of pre-clinical investigation and in early-phase clinical trials. Unfortunately, early evidence has not yielded promising results, but this could be due to the fact that these studies were performed on predominantly molecularly unselected populations. Selecting patients following patient enrichment strategies with a better understanding of the underlying molecular biology, including epigenetic alterations and guided combination approaches, will increase the likelihood of success [33].

### 4.1. RAS/MAPK Signalling Pathway

The mitogen-activated protein kinase (MAPK) signalling pathway is a vital aspect of NSCLC signalling and a predominant aspect in a wide number of cellular functions including cell survival, differentiation, proliferation, metastasis, and apoptosis. Targeting the Ras/Raf/MEK/ERK pathway can prove to be a promising therapeutic regimen for NSCLC patients [34]. Other kinases involved in the Ras/Raf/MEK/ERK cascade include receptor tyrosine kinases (RTKs). Ras/Raf/MEK/ERK are involved in lung cell death, development, and pathogenesis [34]. This family comprises the epidermal growth factor receptor (EGFR) and the fibroblast growth factor receptor (FGFR), which crosstalk with major tumour-promoting signalling pathways. Indeed, when a ligand binds to EGF it stimulates EGFR followed by its activation by an intracellular tyrosine kinase domain, thereafter resulting in its autophosphorylation and its overexpression, causing increased intracellular EGFR pathway activity. This atypical activity might be the reason that almost 40–89% of NSCLC patients have *EGFR* deregulation [34].

MAPK constitutes an evolutionarily preserved family of protein kinases acting as cytoplasmic mediators of signal transduction pathways critical for cellular proliferation and survival. In an LUAD study exploring differences between female and male never smokers, a number of genes relating to the MAPK/PI3K signalling pathway have shown a drastic difference in the prognosis of female and male patients. These include *ERBB4* and *NTF4*, which showed different prognostic effects on LUAD progression in NS males and females. Therefore, sex should be taken into account when designing therapeutics for LUAD never smokers [35].

In a recent study with 83% never smokers, downstream activation of the MAPK signalling correlated with EGFR-pY1197, MAP2K2-pT394, and its substrate MAPK3-pT198/pT202, and in turn with pMAPK1 and other downstream phosphoprotein (RSK2, cPLA2, and STMN1). They observed that the MAPK signalling pathway is commonly activated among both EGFR-WT (wild-type) and mutated patients with different degrees of activation. Indeed, patients without *EGFR*-activating mutations corresponded with low MAPK signalling. Additionally, three of four EGFR-WT samples with higher *MAPK* activity harboured *KRAS* mutations, while samples that harboured both *EGFR* and *TP53* mutations had low MAPK signalling, especially at the later stages. With specific regard to never smokers, variation of MAPK pathway activity was observed with patients having different *EGFR* activating mutations and was also influenced by *TP53* mutations. Lastly, late-stage tumours with lymph-node metastasis seemed to have lower *MAPK* activity [16].

Extracellular regulated kinases (ERK1/2), c-Jun NH2-terminal kinases (JNK), and four P38 enzymes, p38α (MAPK14), p38β (MAPK11), p38γ (MAPK12), and p38δ (MAPK13), are well-characterised cytoplasmic mediators of the MAPK pathway [36]. The expression level of *MAPK11* was found to be significantly higher in an Asian NS cohort [37]. A large number of small-molecule p38 inhibitors have been developed and can theoretically be used to treat tumours that depend on *MAPK14* for progression. This is because *MAPK14* plays a role in cancer cell migration, tumour invasion, and metastasis, where the expression of matrix metalloproteinases (MMPs) and angiogenic factors is induced by p38 MAPK signalling [36]. Activated *ERK* and *JNK* can result in increased proliferation and survival, whereas the P38 MAPK pathway is involved in suppressing tumorigenesis. In NSCLC, *ERK*, *JNK*, and *P38* are usually activated, but their activation degree is variable [25]. 

In an attempt to explore whether the MAPK activation state differs according to smoking status, Mountzios et al. evaluated the expression of activated extracellular signal-regulated kinases including c-Jun and p38 enzymes using immunohistochemistry in LUAD. Following adjustment of any potentially confounding covariates, they found that 37 of 44 NS had higher levels of expression of pP38 compared to 45 of 104 smokers. They observed the P38 pathway to be ten times more activated in never smokers than smokers. Their results provide evidence that life-long non-smoking is associated with an activated P38 pathway and implies that higher *P38* levels are related to distinct molecular changes in never smokers. Since *P38* acts as a tumour suppressor among MAPKs, with higher activation levels observed in never smoker LUAD, this indicates that its action is different in the context of adenocarcinoma cells in never smokers, given their unique molecular and biological characteristics. They indeed confirmed this hypothesis by studying the effects of *P38* pharmacological inhibition on cell growth in the *EGFR* mutant (delE746_A750) adenocarcinoma cell line (HCC827), which is derived from never smokers that do not harbour the *KRAS* mutation. Indeed, *P38* activity contributed to HCC827 cell growth rather than inhibiting it. However, *P38* induced apoptosis or cell senescence in several models characterised by *RAS*-induced proliferation, and contributed to cell growth in LUAD in never smokers. Therefore, it is speculated that the high levels of activated *P38* in never smokers could be explained by the lack of *KRAS* mutations. Specific aberrations in MAPK or interacting pathways responsible for P38 pathway activation in never smoker LUAD are yet to be determined [25].

The expression of ERCC1 protein is a main predictor of the benefit of cisplatin-based chemotherapy in lung cancer, and its gene contains AP-1 sites bound by the transcription factors JUN and ATF2. Planchard et al. investigated whether p38 MAPK activity contributed to excision repair cross-complementation group 1 (ERCC1) mRNA expression and viability of cisplatin in lung cancer cell lines from light or never smokers. They found ERCC1 protein levels to be predicted by activated p38 MAPK in LUAD tissue; furthermore, cells from LUAD light or never smokers generally rely on p38 MAPK signalling for survival, with higher expression of ERCC1 in never smokers. Downregulation of *ERCC1* expression reduced cell viability and could account for the effect of p38 MAPK inhibition on cell viability. Inhibition of p38 MAPK with a specific inhibitor SB202190 that targets both *MAPK11* and *MAPK14* resulted in decreased cell viability in all never smoker cancer cell lines to different degrees. *MAPK11* downregulation reduced cell viability in all cell lines; however, *MAPK14* downregulation also reduced cell viability in a cell line derived from never smokers. This enunciates that *MAPK11* signalling is the main contributor to cancer cell survival in never smokers. *MAPK11* and *MAPK14* have opposite effects on cell differentiation and survival. In *MAPK14* knockout mice, the proliferation of immature lung stem cells also facilitates KRAS G12V tumorigenesis. Furthermore, pre-treatment of two cell lines from never smokers (H1793 and H1651) with SB202190 sensitised cells to cisplatin, which could provide insight into why not all lung cancer patients benefit from cisplatin-based chemotherapy. Additionally, sensitivity to cisplatin was higher following *MAPK11* downregulation of H1651, a cell line from never smokers, or *MAPK14* downregulation of H1650, a cell line from light smokers. This could clarify the cytotoxic effects observed in certain current treatments. Moreover, the crosstalk between p38 and JNK pathways should be investigated as they share many upstream regulators [36].

### 4.2. Mutations in Other EGFR Signalling Pathway Genes

Although mutations in never smokers are lower than in smokers, they are perceived to cause malignant transformation, whereas, in smokers, they are mostly thought to be passenger mutations [21]. *EGFR* mutations were observed in 40–60% of NSCLC NS patients, of which 17% accounted for LUAD [38] and were more common in never smokers or light smokers [39]. Yet, this does not mean that smokers do not have *EGFR* mutations, but that only common *EGFR* mutations were more frequent in never smokers, whereas smokers had more uncommon single and complex rare mutations [40].

A significant revolution in NSCLC therapeutics is the identification of activating oncogenic aberrations such as *EGFR* mutations. *EGFR* tyrosine kinase inhibitors (EGFR-TKIs) are linked with superior efficacy in NSCLC patients with activating *EGFR* mutations [41] and never smokers [42]. NSCLC patients with *EGFR* activating mutations have an excellent response to EGFR-TKIs; however, approximately 20–30% of NSCLC patients with *EGFR* mutations show de novo resistance to EGFR-TKIs. A possible explanation could be the presence of genetic alterations affecting genes downstream of *EGFR*. The preliminary results of a recent clinical study NCT01405079 in NSCLC stage II–IIIa patients with *EGFR* mutation treated with gefitinib versus vinorelbine/platinum indicated that patients on adjuvant gefitinib have a better disease-free survival (DFS) than those on chemotherapy (38.7 months vs. 18 months) [41].

Nevertheless, identifying the *EGFR* mutation is not sufficient to determine the patient’s response to TKIs due to the presence of secondary *EGFR* mutation or any downstream or altered signal activation [41]. Hence, comprehensive genotyping, specifically of interactions with *EGFR* mutation, would be able to provide a better picture of any signal activations that may be present in the patient. In fact, several studies in NSCLC have reported that other signalling pathways mediate potential resistance to EGFR-TKIs; for example, activation of JAK2-related signalling upregulated *ROR1* via NKX2-1, resulting in the overexpression of *NOTCH1*, leading to epithelial-to-mesenchymal transition (EMT). Moreover, EGFR-TKI resistance in patients with T790M mutation resulted from increased DNA repair due to high levels of *BRCA-1*. Additionally, NFKB signalling presented TKI resistance to *EGFR*-mutant NSCLC cells with smoking and never smoking history; however, inactivating *NFKB* using TLR-9 agonist along with erlotinib did not increase PFS in comparison to using erlotinib alone [41]. Indeed EGFR-TKIs have shown better responses in LUAD patients with no smoking history, in the female demographic, of Asian ethnicity, or with *EGFR* mutation [43].

#### 4.2.1. Notch Signalling Pathway Genes

Notch signalling pathway is essential for embryonic lung development and tissue homeostasis. Activating mutations in NSCLC correlate with a worse prognosis [43]. *Notch1* contributes to EGFR-TKI acquired resistance in NSCLC. Moreover, *Notch3* expression is positively correlated with *EGFR* expression, and *Notch3* overexpression is associated with poor prognosis in NSCLC [44].

One study aimed to assess the impact of Notch signalling on survival by examining the expression of *Notch1, 2, 3, 4* in comparison with the adjacent normal tissues in resected NSCLC using RT-PCR. They found a higher expression of *Notch2* in females, as well as never smoker LUAD patients, than tumours of other histologies. They also found the expression of *Notch2* to positively correlate with more advanced lung cancer stages and a higher rate of recurrence or metastasis; ergo, it could be involved in EMT progression [45]. However, this finding was not statistically significant, and its expression had no impact on DFS and OS in LUAD patients. Therefore, *Notch2* signalling plays a crucial role in mutation of LUAD, specifically in never smoker East Asian females. They also found LUAD patients with high expressions of both *Notch1* and *Notch3* to have poor DFS. In addition, Cox regression analysis showed that *Notch3* expression remained the leading predictive factor of DFS [44].

A comprehensive genomic analysis of classic SCC indicated that 25% of human SCCs are affected by genomic alterations of the NOTCH signalling family, and almost 77% of SCLCs with high expression of neuroendocrine markers show a gene expression pattern suggestive of low Notch signalling activity, including a high level of *ASCL1*. Thus, inactivation of both *p53* and *RB1* is critical for tumorigenesis of SCLC, and the inactivation of Notch signalling causes neuroendocrine differentiation. *EGFR* mutation-positive LUAD cells that harbour an inactive form of both *RB1* and *TP53* are more likely to transform [15]. Moreover, the activation status of *Notch1* had a poor prognostic impact on NSCLC, and it was used in the subgroup of p53-negative NSCLC patients to predict overall survival [44].

To identify key genetic changes defining histological transformation from LUAD to SCC, Kabo et al. recruited female never smokers with *EGFR*-mutant LUAD patients. Patients harboured a deletion mutation in exon 19 of *EGFR*. They then compared the gene alteration profile in the original LUAD prior to EGFR-TKI treatment and in the transformed SCC. They ensured that the samples were purely LUAD by surgical resection, and FFPE samples were micro-dissected. They also identified the inactivation of both *RB1* and *TP53* in SCC and LUAD. Additionally, they selected five genes as candidate key genes for the transformation from LUAD to SCC. All three cases had completely matching changes in the nucleotides with additive alterations common in all cases of the five genes identified: *MTOR*, *JAK1*, *NOTCH2*, and *CSF1R*; additionally *MAP2K2* was a lost alteration in all cases. Duplication alterations of *MTOR*, *JAK1*, *NOTCH2*, and *MAP2K2* were located in the 3′ untranslated region (UTR), and the *CSF1R* alteration was a single-nucleotide polymorphism in the intron. There was no common aberration in completely matching nucleotides for *PI3K* and *AKT*. They also focused on *NOTCH* mutations as possible alterations that represent the transformation from LUAD with *EGFR* mutations to SCC, as the expression of *NOTCH* seems to be the key mechanism for the transformation. *NOTCH* mutation and reduced expression at the protein level were detected only following the transformation [15].

The expression levels of *ACSL1* and *MTOR* were higher in LUAD, while those of *NOTCH1/2*, *CSF1R*, and *JAK1* were lower than in transformed SCC. Additionally, *EGFR* expression disappeared in the transformed SCC compared to that in LUAD, and *Rb* expression was not observed in either LUAD or the transformed SCC. They also focused on the expression of *ASCL1*, a downstream gene of Notch signalling regulated through HES-1 and HEY-1, which act as ASCL1 transcriptional repressors. *ASCL1* expression was found to be higher in SCC transformed tissues. *TP53* alteration was detected in the paired samples of both LUAD and transformed SCC in all three cases. Aberrations in *TP53* were detected in all three cases as non-synonymous coding or frame-shift effects, which resulted in amino-acid changes. The authors hypothesised that the inactivation of *p53* and *RB1* emerged during carcinogenesis, and tissues that acquired *Notch* inactivation subsequently transformed, since neuroendocrine differentiation did not occur in tissues lacking *NOTCH* inactivation. *NOTCH2* expression was negative in the transformed SCC, but positive in LUAD. *ASCL1* expression was positive in the transformed SCC but negative in LUAD. They concluded that *NOTCH* mutations were detected as additional alterations in all three cases. It is suggested that *Notch* inactivation is one of the key conditions causing SCC under *RB1* and *p53* inactivation, indicating that the *NOTCH* and *ASCL1*-dependent pathway represents a key process in the transformation using actual tumour tissues from patients with the transformed SCC after becoming EGFR-TKI-resistant. Therefore, the NOTCH/ASCL1 axis could be a potential therapeutic target in transformed SCC from LUAD with oncogenic driver mutation, and AKT inhibitors could delay transformed neuroendocrine lung carcinoma [15].

#### 4.2.2. Abnormalities in Tumour Suppressor Gene Pathways: P53 and KRAS

The incidence of *TP53* mutation is higher in smokers than in never smokers and among patients with SCC compared to LUAD with a different biological impact among the smoking groups. Never smoker lung cancer patients show a totally different and random grouping of *p53* mutations. Moreover, a lower frequency of *TP53* mutations was identified among patients with *EGFR* mutations who were never smokers with LUAD. Additionally, in never smokers, *TP53* mutations were identified as a significant independent negative prognostic factor [46]. In lung cancer, *TP53* mutations are the most prevalent and often co-exist with driver mutations, being higher in SCC than LUAD [47].

A study analysing the association among mutation status, clinicopathologic characteristics, and outcome in never smokers with LUAD identified never smokers to have a higher incidence of targetable mutations with a significantly longer survival than patients without mutations. The authors identified *EGFR* mutations amid the most common encountered mutations—55.6% with deletions in exon 19 and associated with longer OS. They also found the frequency of *ALK* rearrangements (12.3%) to be associated with ipsilateral mediastinal or subcarinal lymph-node metastasis (N2) and a better outcome compared to the wild-type (WT). They also found 14.3% of the tumours to harbour *TP53* mutation. Lastly, they observed significant differences in survival for patients with *EGFR* mutations compared to EGFR-WT (wild type) and *EGFR* pan-negative tumours, as well as *ALK* rearranged versus WT and *ALK* pan-negative tumours [48].

Another study aimed to investigate the impact of concomitant *TP53* mutations and their clinicopathological characteristics in *ALK*-rearranged NSCLC patients, as well as the association of *TP53* with the effect of crizotinib in *ALK*-rearranged patients. In their study, never smokers accounted for 76.6% of the patients and *TP53* mutations occurred in 23.4% of *ALK*-rearranged NSCLC patients. They explored the correlation between *TP53* mutations and the outcome of *ALK*-rearranged patients following crizotinib treatment. They found especially non-disruptive *TP53* mutations to negatively affect the response to crizotinib and correlate with shorter PFS in patients with *ALK*-rearranged NSCLC patients. Non-disruptive *TP53* mutations, which cause partial loss of *p53* function having a retained functional property associated with gain of function (GOF) representing a heterogenous subgroup of *ALK* rearranged NSCLC patients with inferior PFS [47]. These results indicate the negative prognostic role of *TP53* mutations in ALK-rearranged NSCLC patients undergoing treatment with crizotinib.

In addition to the observed higher mutational burden and co-occurring mutations in smokers, they also have a more complex *KRAS* mutation than that observed in never smokers [48], suggesting a different mechanism of carcinogenesis in never smokers. A study analysed *TP53* and *KRAS* mutations in lung cancer tumours of different smoking groups. Their results support the notion that tumorigenesis in lungs proceeds through different molecular mechanisms according to smoking status, and that the accumulation of N-Tyr in never smokers tumour cells is higher than in smokers, implying an aetiology involving severe inflammation. N-Tyr is a stable product of nitration of tyrosine residues and a biomarker of protein damage from peroxynitrite and other reactive nitrogen species (NOS), common in severe forms of inflammatory airway diseases such as chronic obstructive pulmonary disease (COPD) and asthma. However, they did not observe any correlation between N-Tyr and a particular *TP53* mutation type, which could indicate that mechanisms causing severe inflammation other than *TP53* mutation could contribute to carcinogenesis. Nevertheless, they also found that TP53 mutations were detected in 47.5% of never smokers with G:C-to-A:T transitions. It is also possible that the G:C-to-A:T transitions at non-CpG sites in never or former smokers might represent a DNA fingerprint for NNK in patients exposed to secondary smoke. Furthermore, *KRAS* mutations were detected in 15.3% of the cases and were more frequent in LUAD than SCC and in former smokers than in other categories [49]. 

The prognostic impact of *EGFR*, *KRAS*, or *TP53* mutations in LUAD indicated that these genes are not independently associated with prognosis in patients who underwent pulmonary resection, including never smokers. However, using univariate analysis, all except *KRAS* have significant prognostic value, insinuating that the significance is possibly caused by confounding other prognostic factors including sex, smoking status, and tumour differentiation, which, following adjustment, lost their prognostic impact [50].

#### 4.2.3. PI3K–AKT–mTOR

The PI3K–AKT–mTOR pathway is involved in the regulation of several functions including adhesion, motility, invasion, and cell proliferation and differentiation. In NSCLC, abnormal activation of the PI3K–AKT–mTOR pathway seemed to generate resistance to EGFR-TKIs. Alterations can happen through the activation of tyrosine kinase receptors upstream of *PI3K* and *PIK3CA* amplifications, mutations in *KRAS*, *PI3K*, *AKT*, and *TSC1/2*, or loss of *PTEN*. *PIK3CA* and *AKT1* mutations and *PTEN* loss, which are the prominent mutations leading to activation of the PI3K–AKT–mTOR pathway. mTOR inhibitors including everolimus, which targets mTORC1, and temsirolimus are approved for cancer treatment. Moreover, genetic mutations in the PIK3CA/AKT/mTOR pathway, one of the EGFR downstream pathways, might impact the response to EGFR-TKI in NSCLC with activating *EGFR* mutations [51].

Co-occurrence of *PI3K*-related mutations with *EGFR*-activating mutations leads to worse prognosis and shorter PFS with EGFR-TKIs. This is because such alterations may uncouple *EGFR* from downstream signalling. Resistance to EGFR-TKIs would be evident by shorter survival outcomes and/or poor response rates. Patients with *PTEN* mutations had a poor survival outcome, while those with *PIK3CA* or *STK11* mutations revealed trends toward a poor survival outcome [51]. Other proteins related to MAPK/PI3K signalling included *ERBB4* which seemed to show differential effects on LUAD progression in never smokers, with totally different prognostic effects in female never smokers compared to males. It had little effect on the prognosis in the whole LUAD population; however, its level of expression correlated with prognosis in never smokers. *ERBB4*, which is a member of the *EGFR* family, was reported to have abnormal activation via somatic mutations which could associate with tumour progression. Its higher expression in this female LUAD cohort seemed to correlate with poor prognosis in females, but a better prognosis in males [35].

A recent study attempted to explore the genomic characteristics of the PI3K pathway activated in NSCLC patients following progression on EGFR-TKIs and the co-occurrence of common mutations through PI3K–AKT–mTOR. The study was further performed on six patients with a history of everolimus and EGFR-TKI treatment to estimate the anti-tumour activity. These stage IV NSCLC patients had specific mutations along the PI3K–AKT–mTOR pathway, and three of them were never smokers. Following progression on EGFR-TKIs, all patients acquired *PIK3CA* mutations and *PTEN* loss, and they achieved stable disease. Following several patient deaths, the authors inferred that EGFR-TKIs, along with everolimus, might not be enough to overcome EGFR-TKI resistance induced following abnormal PI3K pathway activation. Additionally, PI3K pathway alterations seem to serve as a common resistance mechanism, being present in 14.9% of EGFR-TKI resistance events. Nevertheless, the authors speculated that the failure of therapy was due to the specific targeting of *mTORC1* and not *mTORC2*, since inhibition of *mTORC1* solely can activate an *mTORC1* negative feedback loop, resulting in *AKT* activation via S6K-dependent upregulation of the IRS-1 and TGFR-1 pathways. Therefore, inhibiting *mTORC1* does not completely suppress the PI3K pathway. Moreover, there are other players involved since PI3K pathway activation interacts with other signalling pathways, including the MAPK pathway. Lastly, the safety and toxicity profile of PI3K pathway inhibitors remain unclear and pose issues. Additionally, combination therapy provided limited anti-tumour activity in patients with dysregulated PI3K–AKT–mTOR pathway [52].

Kim and colleagues aimed to investigate the relevance of *EGFR*-downstream gene mutations including *PIK3CA*, *AKT1*, *PTEN*, and *STK11*, and of treatment outcomes of EGFR-TKIs in never smokers with activating *EGFR* mutations. They aimed to prove the concept that the mutation in EGFR downstream genes may be related to EGFR-TKI resistance. Following screening of those patients, the frequency of genetic mutations related to *EGFR* downstream signalling included 3 (4.4%) patients with *PIK3CA* mutation (exons 9 and 20), 11 (16.1%) with *PTEN* mutation (exons 1–9), 4 (5.9%) with *AKT2* mutation, and 9 (13.2%) with *STK11* mutation (exons 1–9). It is of importance to note that the frequency of *PTEN*, which was 16.1% in this NS group, was found to be higher than previously reported studies, with mutations occurring in the phosphatase domain indicating alteration of gene function. They observed the EGFR-TKI treatment outcome of 55 patients including gefitinib (61.8%), erlotinib (36.3%), and a pan-HER inhibitor (1.9%). They found that 20% of the patients showed de novo resistance to EGFR-TKIs, and patients with mutations in the *EGFR* downstream genes had a significantly higher resistance to EGFR-TKIs than those without mutations regarding objective response rate (ORR). Mutations in *PTEN* or *STK11* were significantly associated with a low ORR to EGFR-TKIs; mutations in *PIK3CA* and *AKT* did not differ significantly in terms of treatment response but showed a trend toward poor responses. Of the 55 patients who were treated with EGFR-TKI, the median PFS and OS were 10.3 and 21.2 months, respectively. Patients with mutations in the genes downstream of *EGFR* had significantly shorter median PFS and OS compared to patients without mutations (3.0 vs. 12.0 months, *p* = 0.060; 18.9 vs. 25.0 months, *p* = 0.048). Collectively, their study suggests that the presence of mutations in key *EGFR* downstream genes (*PIK3CA*, *AKT*, *PTEN*, and *STK11*) could affect the treatment outcome [51].

Another study aimed to investigate the association between ETS exposure and *EGFR* mutations in never smoker NSCLC patients. They found ETS exposure to be associated with a lower frequency of *EGFR* mutations with an inverse relationship. They inferred that the history of ETS prior to diagnosis could serve as a negative predictor for *EGFR* mutations in a similar manner to tobacco smoke. Thus, exposure to ETS could result in a similar carcinogenesis mechanism in never smokers to that in smokers. Indeed, genotoxic and epigenetic changes in smokers such as DNA adduct formation and oxidative DNA damage, as well as an increased number of *p53* mutations, including sister chromosome exchange, were also found in never smokers exposed to ETS. This indicates that cumulative ETS exposure is a major risk for the development of lung cancer even in NS [53].

Lastly, a study comparing the immunohistochemical expression of a panel of *EGFR*-related biomarkers in LUAD smokers vs. NS indicated that *EGFR* expression was higher in tumours from smokers, whereas *pAKT* was mainly overexpressed in tumours from never smokers. Biomarkers included *EGFR*, *pAKT*, PTEN, ki-67, p27, and hTERT from 190 patients with completely resected LUAD, of which 43 were NS. The expression patterns of all biomarkers except *PTEN* were different among the smoking groups, which confirmed that specific abnormalities, including changes in *EGFR* signalling pathways, characterise lung carcinogenesis in never smokers. Tumours from never smokers had a higher expression level of *pAKT* than those from smokers. Thus, *pAKT* might be involved in the development of LUAD, and alternative mechanisms rather than *PTEN* inactivation could be responsible for its increased expression [54].

#### 4.2.4. microRNAs

Expression levels of microRNAs (miRNAs) vary in different types of human cancers, with lung cancer being no exception. Additionally, miRNAs have been demonstrated to be diagnostic and prognostic markers in other types of cancer, including lung cancer. A recent study investigated global expression profiles of miRNAs in never smoker and smoker lung cancer patients with *EGFR* mutation versus wild type. The study revealed *EGFR*-mediated regulation of miRNA expression. They employed 29 matched pairs of lung cancer and their adjacent normal tissue from never smokers. The expression of miR-21 was upregulated in smokers in comparison to never smokers with remarkable changes in cases with *EGFR* mutations. The correlation between phosphorylated EGFR (p-EGFR) and miR-21 levels was found to be significant. Furthermore, since miR-21 was found to be suppressed by EGFR-TKI inhibitor, this insinuates that EGFR signalling is a pathway positively regulating miR-21 expression. Moreover, in the never smoker LUAD cell line (H3255) with mutant *EGFR* and high levels of pEGFR and miR-21, anti-sense inhibition of miR-21 enhanced EGFR-TKI-induced apoptosis. With another LUAD cell line (H441), also from never smokers having EGFR wild-type, the anti-sense inhibition of miR-21 not only enhanced the effect of EGFR-TKI, but also induced apoptosis on its own. The expression of miR-21 was identified to be a downstream effector of the activated EGFR signalling pathway with a major oncogenic role in lung carcinogenesis, which could be utilised to improve response to EGFR-TKI therapy [55].

## 5. Application of Molecular Biology and Future Directions

Since LCINS patients harbour significantly lower mutational frequency compared to smokers, with dominance of *EGFR* and *ALK* rearrangements, almost 30% of LCINS cases are driven by genetic mutations treatable with targeted therapies and biomarkers. The vast majority of these alterations tend to be responsive to targeted therapy, curable if caught early, and more preferable than immunotherapy in never smokers including *EGFR* and *ALK* mutations [3]. In fact, there are treatments either available or under investigation in clinical trials targeting the most recurrent genomic alterations including *TP53* or MDM2–TP53 interactions, as well as mutations in *EGFR* or *ERBB2*. A study classifying LCINS identified the RAS pathway to have a distinct impact on survival. Additionally, the authors urged further exploration of tumours with *TP53* and *EGFR* alterations, as well as tumours with loss of chromosomes 22q and 15q or CHEK2 LOH, as they could be promising therapeutics. This is because these five independent genomic alterations presented a twofold higher mortality rate, which insinuates that compounds targeting bystander genes that are deleted along with tumour suppressor genes in chromosome arm losses should be further investigated [1]. 

Unfortunately, immunotherapy has not had a huge impact on LCINS treatment because the immune landscape in never smokers has not been much explored, and immune target proteins including PD-L1 are more common in smokers. Moreover, immunotherapy might not be beneficial for tumours with a scarcity of driver mutations and low TMB, which have limited targets for therapeutic intervention. Instead, this subtype might benefit from targeting *KRAS-* and stem-cell-associated signalling pathways. Nevertheless, to be able to identify these genetic alterations and key mutations, patients must have a high-quality tumour biopsy with adequate tumour cellularity to allow clinical genomic testing. Govindan and colleagues were able to identify targetable mutations in 80% of never smoker samples using whole-exome sequencing (WES) and targeted deep sequencing. The study also confirmed the sex bias, with females being overrepresented in the never smoker cohort. Intriguingly, Govindan et al. found the overall prevalence of germline alterations to be comparable among the smoking groups; thus, ETS, as inferred from the mutational signature analysis, is also likely to predispose to LCINS. Their results, therefore, highlight the need to classify never smoker tumours into immune subtypes and the applicability of multi-omics approaches involving tumour tissue and germline DNA analyses [3].

Approved EGFR-TKIs for NSCLC patients with locally advanced or metastatic NSCLCs with activating TK mutations include erlotinib and gefitinib. Several reports have provided evidence of the efficacy of TKIs such as erlotinib and gefitinib for *EGFR* mutation-positive cancers and crizotinib for tumours harbouring *ALK* rearrangements. The median PFS in mutation-positive patients receiving erlotinib was 9.7 months, whereas it was 5.2 months in those receiving standard chemotherapy. Moreover, clinical trials in *EGFR* mutation-positive tumours for gefitinib and erlotinib showed a higher response rate of 68% compared to 8–9% in mutation-positive tumours lacking *EGFR* mutation. This is even more proof that molecular profiling of advanced NSCLC to select LUAD patients for targeted TKI therapy improves survival [23].

Another study to identify tumorigenesis hallmarks and druggable targets included early-stage LUAD female NS patients with *EGFR* mutations. The authors aimed to elucidate driver mutations that predict therapeutic efficacy. They evaluated whether patients with EGFR-L858R had different outcomes compared to EGFR-Del19 patients in an independent retrospective cohort including treatment-naïve, completely resected pathologic stage IA and IB patients. Stage IA showed no difference in OS; however, different outcomes were observed for stage IB patients with EGFR-L858R, which had a significantly lower OS compared to Del19 patients. This indeed confirms the likely divergence and a higher tendency of cancer metastasis at a later stage, and it shows that key genomic features can alter the proteomic taxonomy of specific tumours. This proteomics-based classification could help develop strategies for the management of early-stage NS LUAD [16] and highlight the need for more personalised in-depth studies.

## 6. Conclusions

Ongoing efforts to unravel the genomic and epigenomic underpinnings of LCINS will allow further development of novel highly selective targeted therapies, ultimately improving survival outcomes for NS patients. Several genome-wide association studies have mostly included predominantly smokers, with limited analysis of NS or even ETS. Notably, as the incidence of lung cancer is increasing in several developed countries, it is essential for future studies to focus on NS, as well as include information regarding whether the patients were frequently exposed to passive smoke or other pollutants. Other risk factors clearly have implications for lung carcinogenesis in NS, and it is necessary to improve knowledge with regard to other contributors. Genetic studies suggest that lung cancer has distinct aetiology and progression, at least at the molecular level, in never smokers. Moreover, LCINS tumours differ distinctly in terms of molecular pathology and response to treatment from smoking-associated lung cancer. LCINS patients with adenocarcinoma frequently carry mutations within the tyrosine kinase domain of the *EGFR* gene, whereas smokers tend to have *KRAS* mutations and are associated with resistance to EGFR-TKI, thus emphasising the need for classification of mutational status, prior to targeted therapy studies and clinical trials, to provide guidance for the preferred treatment route. Ergo, genetic alterations and patterns of mutations specific to LCINS should be explored. This will aid in providing optimal treatment approaches for LCINS. We anticipate future analyses to include prospective data collection, especially regarding mutation status, smoking status, and risk factor exposures of LCINS patients. This will reveal additional therapeutic targets of relevance for a majority of lung cancer patients.

## Figures and Tables

**Table 1 ijms-24-13314-t001:** Comparison of mutation frequency of LUAD in never smokers (NS), current smokers (CS), and reformed smokers (RS). *p*-Values were calculated using the chi-square test. Statistically significant (*p* < 0.05) results are marked with an asterisk (*).

Gene	Mutated Samples	Samples Tested	NS %	CS %	RS %	*p*-Value
*EGFR **	25,394	95,066	13.33	1.67	6.60	*p*-value = 0.0046
*KRAS*	5994	37,187	6.67	14.17	16.50	*p*-value = 0.095
*TP53*	3246	9440	10.67	24.17	21.78	*p*-value = 0.058
*PIK3CA*	546	13,654	1.33	3.33	2.97	*p*-value = 0.800
*CSMD3*	543	2305	1.33	7.50	9.57	*p*-value = 0.059
*LRP1B **	497	2492	1.33	15.83	15.18	*p*-value = 0.0042
*BRAF*	493	24,447	0.00	5.00	4.62	*p*-value = 0.14
*USH2A **	484	2305	5.33	10.83	15.84	*p*-value = 0.038
*RYR2 **	444	2199	6.67	21.67	16.17	*p*-value = 0.021
*CDKN2A*	425	5443	0.00	2.50	1.98	*p*-value = 0.45
*MUC16 **	402	2333	4.00	16.67	21.45	*p*-value = 0.0018
*ZFHX4 **	379	2200	1.33	15.00	13.86	*p*-value = 0.0071
*SPTA1*	308	2202	6.67	10.00	13.20	*p*-value = 0.24
*STK11*	298	5856	2.67	5.83	9.90	*p*-value = 0.073
*MUC17*	288	2205	4.00	12.50	8.91	*p*-value = 0.13
*SYNE1*	287	2204	0.00	5.83	3.63	*p*-value = 0.072
*XIRP2*	283	2200	0.00	10.83	10.23	*p*-value = 0.014
*FLG*	282	2203	5.33	15.00	12.54	*p*-value = 0.12
*ERBB2*	275	18,935	4.00	0.00	0.99	*p*-value = 0.054
*FAM135B **	256	2303	0.00	5.00	7.92	*p*-value = 0.015
*COL11A1*	255	2302	1.33	8.33	9.57	*p*-value = 0.063
*NAV3 **	253	2203	0.00	12.50	9.24	*p*-value = 0.0086
*RYR3*	246	2199	1.33	5.00	6.93	*p*-value = 0.16
*KEAP1*	240	2993	2.67	6.67	9.57	*p*-value = 0.12

**Table 2 ijms-24-13314-t002:** A list of the identified alterations in never smoker NSCLC patients discussed in this paper.

Gene	Pathway	Findings	References
*TP53*	Notch/ASCL1 axis	Inactivation of *TP53* and *RB1* participated in the transformation of SCLC in female never smokers.	[15]
*RB1*			[15]
*UBA1*	Ubiquitin pathways	Mutations in *UBA1* occurred before the corresponding copy number gain.	[1]
*ARID1A*	PI3K/AKT pathway	Mutations in *ARID1A* could promote exit from a quiescent cell state, resulting in high intra-tumoural heterogeneity.	[1]
*CTNNB1*	WNT pathway	Never smokers had a lower TMBs and a higher frequency of mutations in *CTNNB1* compared with smokers.	
*BRCA1*	DNA repair	Germline alterations in the listed DNA repair genes were exclusively mutated in never smokers.	[3]
*BRCA2*			
*FANCG*			
*FANCM*			
*MSH6*			
*POLD1*			
*RNF213*	WNT pathway	Somatic mutations were identified as prevalent in Taiwanese study of never smoker lung adenocarcinoma patients.	[16]
*ATP2B3*			[16]
*TET2*			[16]
**EGFR-Related Mutations**			
*KRAS*	RTK–Ras pathway	*EGFR* and *KRAS* mutations were mutually exclusive (*p* = 0.004)	[1]
		Mutations in *KRAS* were generally early events occurring prior to whole-genome doubling and most other somatic copy number alterations.	[1]
*ALK*		6.0% alteration in RTK–Ras pathway.	[1]
		Never smoker lung adenocarcinomas had a higher proportion of *ALK* mutations (12%) than former/current smokers (2%) (*p* < 0.0001).	[17]
*MET*		4.3% alteration in RTK–Ras pathway.	[1]
*ERBB2*		3.9%, all indels alteration in RTK–Ras pathway.	[1]
*ROS1*		2.6% alteration in RTK–Ras pathway.	[1]
*RET*		1.3% alteration in RTK–Ras pathway.	[1]
*EGFR*		Higher frequency of *EGFR* mutations in female (31.4%) than in male patients (19.3%) (*p* = 0.092).	[1]
		Somatic mutations occurred in 85% of Taiwanese never smoker lung adenocarcinoma patients.	[16]
		Statistically significant between NS and CS LUAD (*p* < 0.005).	[9]
		Mutations in *EGFR* were generally early events occurring prior to whole-genome doubling and most other somatic copy number alterations.	[1]
		Never smoker lung adenocarcinomas had a higher proportion of *EGFR* mutations 37% than former/current smokers 14% (*p* < 0.0001).	[17]
		Erα expression was correlated with the presence of an *EGFR* mutation in female never smokers.	[18]
*RBM10*		Mutations in *RBM10* were generally early events occurring prior to whole-genome doubling and most other somatic copy number alterations.	[1]
		Top ranking mutations in 20% of Taiwanese never smoker lung adenocarcinoma patients.	[16]
*TP53*		Top-ranking mutations in 33% of Taiwanese never smoker lung adenocarcinoma patients.	[1]
		Mutations in *TP53* were generally early events occurring prior to whole-genome doubling and most other somatic copy number alterations.	[1]
*SETD2*		Significant enrichment of *SETD2* mutations in samples with oncogene fusions, particularly in *TP53*-proficient tumours (*p* = 0.06).	[1]

## Data Availability

Not applicable.

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
