# Peer review of "A Functional Genomics Review of Non-Small-Cell Lung Cancer in Never Smokers"

_ijms, 2023, doi:10.3390/ijms241713314_

Round 1
Reviewer 1 Report
1. Authors explained the study well. Are there any epigenetic pathways involved? Please include the information regarding the epigenetic pathway.
Minor English editing is required. (Grammer)
Author Response
Thank you very much for your kind words, we appreciate the time and effort that you have dedicated to providing your valuable feedback on our manuscript. Also, it really is encouraging to hear that the study is explained well.
We agree epigenetics is an essential part of cancer development and needs to be further explored. We mentioned methylation profiles briefly (lines 298-299) and epigenetic changes (lines 832-836). Indeed, it would have been interesting to explore this aspect. However, in the case of our review it would be out of scope since we based our review on pre-clinical and clinical trials, there aren’t many studies to cite and build a bigger comprehensive pathway picture, thus we included epigenomics rather than epigenetics of never smokers in NSCLC.
Therefore, we tried to include a component of the epigenetic machinery that is non-coding RNAs section 4.2.4 (lines 734-751).
Additionally, regarding epigenetic pathways in never smoker lung cancer patients, more precisely NSCLC, studies focused solely on never smokers are recently emerging, as I am certain you are aware, it is usually more investigated in relation to smoking related DNA methylation or radon exposure and for instance CpG methylation changes associated with folic acids deficiency which are usually are also investigated in association with smoking.
Moreover, the findings across research papers-not clinical/pre-clinical are not very consistent and usually include co-morbidities such as COPD (10.1186/s12931-019-1222-8,10.1136/bmjresp-2014-000032), or with insignificant changes in the expression of homeobox-related genes(10.1016/j.cancergen.2014.12.002) or are discussed in relation to mutation.
Reviewer 2 Report
In the review paper entitled “A Functional Genomics Review of Non-Small Cell Lung Cancer in Never Smokers,” the authors focus on lung cancer in never smokers (LCINS), examining the distinctions between LCINS and lung cancer in smokers. They provide a comprehensive summary of the genetic, genomic, and epigenomic characteristics of LCINS, including differences in somatic mutations, tumor mutational burden, and chromosome aberrations. Particular attention is paid to specific genetic features like EGFR tyrosine kinase mutations and EML4-ALK gene rearrangements, and their associations with chemotherapy resistance.
It would be appreciated if the authors could summarize the main findings listed in this review paper into one or two figures. Doing so would visually represent the complex data and relationships explored in the text, providing readers with an easily digestible overview. These visual aids could emphasize key differences, correlations, or trends, and may include charts, graphs, or diagrams to effectively convey the most crucial aspects of the research. By consolidating this information into a visual format, readers could more quickly grasp the central themes and conclusions of the paper, enhancing the accessibility and impact of the work.
There are several grammar, spelling, and writing errors in this manuscript; the authors should carefully revise it. For example,
Line 39, remove the first “The” of this sentence.
Line 40: Replace the semicolon after "decreasing" with a comma, and change "amid" to "among."
Line 42: There seems to be a verb missing after "never smokers.". "... yet approximately 10-25% of lung cancer cases occur in never smokers."
Line 54: It might be more clear to add "usage" or "popularity" to the last sentence. This sentence should be corrected to: "This may have been due to an increase in smoking among women, especially the trendy light flavored cigarettes' popularity."
The language should be remarkbly improved.
Author Response
We appreciate the time and effort that you have dedicated to providing your valuable feedback on our manuscript. We are grateful to your insightful comments on our paper and have been able to incorporate changes to reflect most of the suggestions provided and hope the changes are satisfactory.
We also agree with your comment regarding the visual aid and have added the suggested figure as a graphical abstract which complements Table 2.
We would also like to apologise for the spelling and grammatical errors. We have accordingly corrected them-including the ones you pointed out and consulted a member of the author proofreading services at our University. They have been updated and can be viewed as tracked changes within the manuscript.
Reviewer 3 Report
Manuscript "A Functional Genomics Review of Non-Small Cell LungCancerIn3 Never Smokers" reviews the genetic aspects of non-small cell lung carcinoma, written in great detail. This work will be useful to oncologists, geneticists and pulmonologists in their work. The paper considers in detail almost all currently known genes responsible for cell growth and proliferation of epitheial and glandular tissues. The review may be accepted as presented.
Author Response
We appreciate the time and effort that you have dedicated to providing your valuable feedback on our manuscript.
Thank you for your kind words, it is really encouraging to hear that! We do hope in the future that a few or all of the mentioned genes in this review would be researched and developed in cancer therapeutics or act as non-invasive biomarkers to reduce the devastating global burden of this disease.
Round 2
Reviewer 2 Report
Thank you to the author for addressing all of my concerns and comments. The work is now suitable for the journal.